# Mesenchymal Stem Cell-Derived Extracellular Vesicles: An Emerging Diagnostic and Therapeutic Biomolecules for Neurodegenerative Disabilities

**DOI:** 10.3390/biom13081250

**Published:** 2023-08-16

**Authors:** Mahmoud Kandeel, Mohamed A. Morsy, Khalid M. Alkhodair, Sameer Alhojaily

**Affiliations:** 1Department of Biomedical Sciences, College of Veterinary Medicine, King Faisal University, Al-Ahsa 31982, Saudi Arabia; salhojaily@kfu.edu.sa; 2Department of Pharmacology, Faculty of Veterinary Medicine, Kafrelsheikh University, Kafrelsheikh 33516, Egypt; 3Department of Pharmaceutical Sciences, College of Clinical Pharmacy, King Faisal University, Al-Ahsa 31982, Saudi Arabia; momorsy@kfu.edu.sa; 4Department of Pharmacology, Faculty of Medicine, Minia University, El-Minia 61511, Egypt; 5Department of Anatomy, College of Veterinary Medicine, King Faisal University, Al-Ahsa 31982, Saudi Arabia; kalkhodair@kfu.edu.sa

**Keywords:** mesenchymal cell, extracellular vesicles, neurodegenerative diseases, treatment, MSCs-based therapy

## Abstract

Mesenchymal stem cells (MSCs) are a type of versatile adult stem cells present in various organs. These cells give rise to extracellular vesicles (EVs) containing a diverse array of biologically active elements, making them a promising approach for therapeutics and diagnostics. This article examines the potential therapeutic applications of MSC-derived EVs in addressing neurodegenerative disorders such as Alzheimer’s disease (AD), multiple sclerosis (MS), Parkinson’s disease (PD), amyotrophic lateral sclerosis (ALS), and Huntington’s disease (HD). Furthermore, the present state-of-the-art for MSC-EV-based therapy in AD, HD, PD, ALS, and MS is discussed. Significant progress has been made in understanding the etiology and potential treatments for a range of neurodegenerative diseases (NDs) over the last few decades. The contents of EVs are carried across cells for intercellular contact, which often results in the control of the recipient cell’s homeostasis. Since EVs represent the therapeutically beneficial cargo of parent cells and are devoid of many ethical problems connected with cell-based treatments, they offer a viable cell-free therapy alternative for tissue regeneration and repair. Developing innovative EV-dependent medicines has proven difficult due to the lack of standardized procedures in EV extraction processes as well as their pharmacological characteristics and mechanisms of action. However, recent biotechnology and engineering research has greatly enhanced the content and applicability of MSC-EVs.

## 1. Introduction

Mesenchymal stem cells (MSCs) are a type of adult stem cell with the ability to develop into different types of mesoderm-derived cells. MSCs can be located in various tissues like bone marrow, adipose tissue, umbilical cord, dental tissue, connective tissues of muscle and skin, and endometrial polyps [1,2]. Traditionally, MSCs were not considered to be naturally present in the brain; however, recent studies have indicated that they might exist as perivascular cells in nearly all adult tissues, including the brain [3]. MSCs can be categorized into two main groups, namely embryonic or adult, depending on the specific developmental phase from which they are derived [4]. These cells possess two crucial characteristics: the capacity to transform into different cell lineages and the ability to renew themselves. As multipotent stem cells, they show great potential in preclinical investigations for treating diverse medical ailments [5].

MSCs naturally produce a variety of active ingredients with trophic, immunomodulatory, pro-regenerative, anti-inflammatory, pro-angiogenic, and anti-apoptotic properties [6]. MSCs regulate immune function in a paracrine manner by producing immunomodulatory factors such as transforming growth factor (TGF), hepatic growth factor (HGF), interleukin (IL)6, and IL-10, as well as pro-angiogenic factors like basic fibroblast growth factor (bFGF) and placental growth factor (PGF) [7,8].

MSCs have become a promising therapeutic option for treating neurological diseases due to their distinctive characteristics [9]. These cells are plentiful and easily obtainable from various tissues, and they exhibit low immunogenicity, allowing for allogeneic transplantation without triggering significant immune responses. Additionally, MSCs possess potent anti-inflammatory and immunomodulatory properties, which make them well-suited for addressing neuroinflammatory conditions [10]. Through the secretion of trophic factors [11], they actively promote tissue repair, neurogenesis, and neuronal survival. Their unique ability to cross the blood–brain barrier facilitates delivery to the central nervous system. Notably, clinical studies have demonstrated that MSC transplantation is well-tolerated and associated with low adverse effects [12].

Extracellular vesicles (EVs), which are released by cells, have recently been revealed to play an important role in cell-to-cell communication. Exosomes, microvesicles (MVs), and apoptotic bodies (APBs) are all examples of EVs [13]. MSC-derived EVs (MSC-EVs) have been found to provide therapeutic benefits comparable to their whole-MSC counterparts, demonstrating that EVs are essential mediators of MSC treatment efficacy [14]. These nano-sized vesicles have sparked optimism in curing intractable neurodegenerative illnesses owing to their ability to carry a broad range of biomolecules across a long distance among cells and to overcome biological barriers [15,16] (Figure 1).

The term “neurological disability” refers to a variety of conditions that affect the brain, spinal cord, and nerves throughout the body. Structure, metabolic, or electrical anomalies in the nervous system can cause these illnesses [17]. Neurological disorders can lead to various symptoms, such as paralysis, muscle weakness, impaired coordination, diminished sensation, seizures, cognitive confusion, pain, altered levels of consciousness, and other related manifestations [18]. While certain neurological conditions may be mild and temporary, others are more severe and may necessitate continuous or immediate medical intervention. Behavioral or cognitive disabilities, as well as cognitive, thinking, and reasoning deficits, are symptoms of neurocognitive disorders caused by gradual loss and degeneration of the brain [19]. Neurodegenerative diseases (NDs) are a set of neurological disorders marked by nervous system malfunction and gradual neuronal loss that are linked to aging. Alzheimer’s disease (AD), Parkinson’s disease (PD), amyotrophic lateral sclerosis (ALS), multiple sclerosis (MS), and Huntington’s disease (HD) are the most frequent NDs. These disorders progress over time and are usually accompanied by the buildup of protein aggregation, the composition of which varies depending on the condition [20].

Cognitive impairment affects up to 50% of all MS patients and up to 20% of all PD patients [21,22]. AD makes a considerable contribution to neurological disability because it impairs thinking skills, memory, judgment, and decision-making [23].

The mechanisms of neurodegeneration and the ensuing neurological impairment differ amongst different disorders, yet several characteristics are shared. Abnormal activation of programmed cell death (PCD) pathways, for example, is a typical hallmark of neurodegenerative illnesses, leading to unintended loss of neuronal cells and function [24]. Axonal dysfunction and degeneration play significant roles in the advancement of diseases such as ALS, MS, PD, and HD. They are acknowledged as prominent factors contributing to the progression of these conditions [25]. This review examines various current factors related to the therapeutic capabilities of MSC-EVs in addressing the treatment of AD, HD, PD, ALS, and MS.

## 2. EVs Types and Biogenesis

EVs are nanoscale vesicles with different forms, sizes, contents, and surface marks that are released into the extracellular environment via a variety of methods [26]. These vesicles convey cytoplasm and cell membrane components that have been placed inside them specially. They are produced by every living cell and perform critical roles in a wide range of physiological and pathological processes [27]. Exosomes, MVs, and ABs are all examples of EVs, according to the International Society of Extracellular Vesicles (ISEV) [13]. This classification is based on their diameter size and origin; hence, each subclass of EVs reflects distinct physicochemical features that play critical roles in both normal and pathological circumstances [28].

The MVs, also known as ectosomes, are the first type of vesicle formed by the direct budding of vesicles from the cell membrane to the cell’s exterior [29]. Exosomes are the second type, which form by budding into endosomes to produce multivesicular bodies (MVBs). These MVBs either bind to lysosomes, where the content is digested, or they bind to the cell membrane, where it is ejected as exosomes [30]. Exosomes are 50 to 150 nm in diameter, while MVs are 100 to 1000 nm in diameter [31].

Apoptotic bodies (APBs), the third kind, are formed when cells undergo apoptosis. APBs are formed by the separation of membrane blebs or by the formation of apoptopodia during the apoptotic process. Beaded apoptopodia and non-beaded apoptopodia are the two forms of apoptopodia [32]. APBs have a significantly greater range of diameters, ranging from 50 to 5000 nm [33] (Table 1).

The standardization and extraction of MSC-EVs revealed various characteristics [34]. The most popular isolation methods for MSC-EVs were based on the size of the EV and included ultracentrifugation, differential ultracentrifugation, density gradient ultracentrifugation, and ultrafiltration [35]. A larger yield is obtained while maintaining the biophysical and functional characteristics of the EV isolated from stem cell culture using ultrafiltration followed by size exclusion chromatography (SEC) [36]. Other methods were also used, comprising immune affinity capture. However, it was less efficient than SEC [37].

Extracting EVs from MSC cultures is a critical process in biomedical research, and it requires meticulous attention to multiple precautions to ensure the reliability and quality of outcomes [38,39]. Maintaining a sterile environment throughout the culture and isolation process is crucial to prevent contamination, while the quality and characteristics of the MSCs used can significantly influence the composition and function of the isolated EVs [40]. Therefore, it is essential to utilize high-quality MSCs that meet specific criteria. Additionally, proper handling of samples is of utmost importance to preserve the integrity and stability of EVs. This involves careful collection, storage, and transportation to minimize degradation or loss [41]. To enhance the yield and purity of EVs, optimizing the isolation method is necessary, which may include adjusting centrifugation parameters based on study requirements and EV characteristics [42]. Lastly, to ensure the quality and purity of the isolated EVs, thorough characterization using techniques like electron microscopy, nanoparticle tracking analysis, and protein analysis is indispensable [42,43]. These measures collectively contribute to the success of MSC culture and EV isolation, advancing the understanding and potential applications of EVs in various biomedical fields with reliable and meaningful results.

## 3. EVs Characteristics

Cells produce lipid-bound vesicles in the extracellular environment, which are referred to as EVs [44,45]. MicroRNAs (miRNAs), mRNAs, circular RNAs (circRNA), long noncoding RNAs (lncRNA), proteins, lipids, and metabolites are all possible components of EVs [46,47] (Figure 2). APBs also include entire cell organelles such as mitochondria, endoplasmic reticulum fragments, and ribosomes [33].

### 3.1. Exosome

Exosomes, referred to as ILVs (Intraluminal Vesicles), are discharged by various cell types and have been observed in multiple bodily fluids such as urine, plasma, sperm, bronchial fluid, saliva, cerebral spinal fluid (CSF), serum, amniotic fluid, breast milk, tears, stomach acid, synovial fluid, lymph, and bile [48,49,50,51,52]. Exosomes include a diverse range of lipid-anchored membrane proteins, transmembrane proteins, exosome lumen soluble proteins, and peripherally associated membrane proteins [53,54].

Exosomes have a role in cell survival, cell–cell communication, and tumor growth. They have been shown to promote immunological responses by serving as antigen-presenting vesicles [55,56]. Exosomes have been reported to aid in neurite growth, production of myelin, and neuronal survival in the nervous system, hence aiding in regeneration and tissue repair [57,58].

### 3.2. Microvesicles

The MVs (also known as microparticles or ectosomes) are EVs that are discharged from cell membranes [45]. Megakaryocytes, the placenta, monocytes, blood platelets, tumor cells, and neutrophils are all sources of MVs [59]. They can be detected in both tissues and bodily fluids [60].

The MVs help cells communicate with one another by transporting chemicals, proteins, miRNA, and mRNA between cells [61]. MVs, which were once discarded as cellular trash, may now reflect the antigenic composition of the cell of origin and play a role in cell signaling. Tumor immune suppression, anti-tumor effects, metastasis, angiogenesis, tissue regeneration, and tumor-stroma interactions have all been linked to EVs [62,63,64]. MVs can also help cells get rid of misfolded proteins, cytotoxic chemicals, and metabolic waste. Variations in MVs levels can signal a variety of illnesses, including cancer [65,66].

### 3.3. Apoptotic Bodies

APBs are created exclusively when programmed cell death takes place [67,68]. Smaller vesicles are indeed produced during this activity, which are distinguished by the existence of organelles inside the vesicles [69]. MVs and exosomes have a completely different makeup than APBs. Unlike MVs and exosomes, ABs include intact organelles, a minor quantity of glycosylated proteins, and chromatin [31,70]. As a result, greater amounts of proteins linked with the mitochondria (HSP60), nucleus (histones), Golgi apparatus, and endoplasmic reticulum (i.e., GRP78) are anticipated. Furthermore, the proteome profiles of cell lysate and APBs are quite similar, but the proteomic patterns of cell lysate and exosomes are very different [71,72]. Phosphatidylserine (PS) is indeed the sole ApoBD marker that has been found thus far [32].

## 4. MSC-EVs Components

### 4.1. Pro-Angiogenic Factors

Angiogenesis refers to the formation of new capillaries from previously existing blood vessels [73]. MSC-EVs are both anti- and pro-angiogenic [74] (Figure 3). Vascular endothelial growth factor (VEGF), TGF-β, tumor necrosis factor-alpha (TNF-α), fibroblast growth factor (FGF), and angiopoietins are some of the most well-known angiogenic growth factors and cytokines. Endothelial cells, fibroblasts, smooth muscle cells, platelets, inflammatory cells, and cancer cells are all the main sources of these growth factors [75].

### 4.2. Immunomodulatory Factors

MSC-EVs have immunomodulatory activity and contain chemicals that affect immune cells. They include inflammatory cytokines and chemokines that can influence both innate (e.g., macrophages, dendritic cells, and natural killer (NK) cells) and adaptive immune cells (T and B cells) [76,77] (Figure 3). Overexpression and knockdown procedures were employed by several research groups to discover the active immunomodulatory molecules essential for MSC-EVs’ treatment efficacy [78,79,80]. In their study involving pigs with renal artery stenosis (RAS), Eirin and colleagues observed that the administration of MSC-EVs containing IL-10 resulted in enhanced renal structure and function. Additionally, the treatment led to a decrease in renal inflammation and an increase in the population of reparative macrophages. These benefits were completely lost when EVs generated from MSCs with knocked-down IL-10 were used to treat RAS [81].

The administration of MSC-EVs, which are capable of releasing the TSG-6 protein—a protein that regulates the immune response and is produced during pathological conditions in response to heightened inflammation—proved effective in reducing lung inflammation and cell death in animals with bronchopulmonary dysplasia. However, the therapeutic benefits were not observed when MSC-EVs lacking TSG-6 expression were utilized [82].

MSC-EVs harboring the C-C motif chemokine receptor-2 (CCR2) might prevent monocyte and macrophage activity and protect mice from renal/ischemia harm. Several studies have also shown that particular miRNAs that are overexpressed in MSC-EVs play an immunomodulatory role [83]. In this regard, MSC-EVs containing miRNAs such as miR-142-3p, miR-21-5p, miR-126-3p, and miR-223-3p are endowed with the capacity to regulate dendritic cell maturation and boost their anti-inflammatory activity [84]. Similarly, within a mouse model of polymicrobial sepsis, MSC-EVs containing miR-223-3p demonstrated the ability to reduce the release of pro-inflammatory cytokines in macrophages, suppress the systemic inflammatory response, mitigate heart failure, and enhance overall survival [85]. Conversely, these beneficial therapeutic outcomes were absent when using MSC-EVs obtained from miR-223 knockout mice [85].

MSC-EVs expressing miR-223, on the other hand, have been demonstrated to protect against liver impairment in autoimmune hepatitis types and to down-regulate several cytokines and inflammatory genes [84,86]. MSC-EVs harboring miRNAs, including miR-146a, miR-21a-5p, miR-223, and miR-199a, regulated several inflammatory genes (e.g., NLRP3 and IL-6) and promoted macrophage polarity toward the anti-inflammatory M2 phenotype [87].

The miRNAs contained in MSC-EVs have been shown to play a vital role in the therapeutic function of MSC-EVs in a number of diseases. miRNAs have a vital role in controlling several infectious and non-infectious diseases by regulating gene expression [88]. They have immense potential and practical value in treating several diseases, including neurodegenerative disorders. Table 2 provides a comprehensive overview of various miRNAs found in MSC-EVs, along with their roles and potential applications. These miRNAs demonstrate diverse functions and exhibit promising therapeutic potential in different areas of regenerative medicine and disease treatment. One notable miRNA is miR-22, which inhibits the inflammatory response and contributes to nerve function recovery [89,90]. It has the potential to reduce the release of inflammatory factors and minimize infarct size, suggesting its application in conditions associated with inflammation and tissue damage. Another miRNA, miR-21, plays a crucial role in promoting cell survival and proliferation. Its potential applications lie in tissue regeneration and wound healing, indicating its therapeutic importance in the field of regenerative medicine [89,90].

MiR-let-7, a regulator of cell proliferation and differentiation, shows promise in tissue regeneration and angiogenesis [91]. This miRNA holds potential for therapeutic interventions aimed at promoting tissue repair and vascularization. Furthermore, miR-29b-3p is involved in modulating extracellular matrix remodeling [92]. Its potential applications encompass fibrosis treatment, tissue repair, and fracture healing, highlighting its importance in addressing conditions characterized by abnormal tissue remodeling.

MiR-126 demonstrates a role in promoting angiogenesis and neurogenesis. This miRNA holds promise for cardiovascular regeneration and potential therapeutic applications in neurological disorders [93]. Additionally, miR-133 exhibits anti-inflammatory properties and has the potential to reverse liver injury. Its application may be targeted toward liver injury treatment [86]. MiR-146a-5p [94] and miR-155 [95] are involved in modulating immune responses and inflammation. These miRNAs have the potential for application in autoimmune diseases and inflammatory modulation therapies.

MiR-210 induces vascularization and promotes cellular adaptation to hypoxia. It holds promise for the treatment of ischemic diseases, tissue repair, bone regeneration, and selective regeneration of ischemic heart tissues [96,97]. MiR-223 regulates neuronal cell apoptosis and immune cell reactions, presenting potential applications in conditions such as ischemic kidney and myocardial infarction [98,99,100]. Moreover, the miR-17-92 cluster is associated with increased neural plasticity and functional recovery, particularly in treating stroke [101].

These findings on miRNAs found in MSC-EVs provide valuable insights into their roles and potential applications in various therapeutic areas, paving the way for further research and development in regenerative medicine and disease treatment.

**Table 2 biomolecules-13-01250-t002:** The miRNAs found in MSC-EVs, their role, and their applications.

miRNA	Role/Function in MSC-EVs	Potential Applications	Reference
miR-22	inhibits the inflammatory response and nerve function recovery	inhibit the release of inflammatory factors, reduction of infarct size	[89,90]
miR-21	Promotes cell survival and proliferation	Tissue regeneration, wound healing	[102]
miR-let-7	Regulates cell proliferation and differentiation	tissue regeneration, angiogenesis	[91]
miR-29b-3p	Modulates extracellular matrix remodeling	Fibrosis treatment, tissue repair, and fracture healing	[92]
miR-126	Promotes angiogenesis and neurogenesis	Cardiovascular regeneration, neurological disorders	[93]
miR-133	Reduced inflammation and reversed liver injury	Liver injury	[86]
miR-146a-5p	Modulates immune responses and inflammation	Autoimmune diseases, inflammation modulation	[94]
miR-155	Modulates immune responses and inflammation	Immunomodulation, inflammatory disease therapy	[95]
miR-210	Induce vascularization, Promotes cellular adaptation to hypoxia	Ischemic diseases, tissue repair, bone regeneration, selective regeneration of ischemic heart	[96,97]
miR-223	Regulates neuronal cell apoptosis and immune cells reactions	Ischemic kidney, myocardial infarction	[98,99,100]
miR-335	Promotes osteoblast differentiation	bone fracture recovery	[103]
miR-486	Promotes cardiac regeneration and repair	Heart disease treatment, cardiac tissue repair	[104]
miR-499	Inhibit endometrial cancer growth and metastasis	anticancer	[105]
miR-17-92 cluster	Increases neural plasticity an functional recovery	In treating stroke	[101]

### 4.3. Anti-Apoptotic Factors

MSCs have anti-apoptotic properties [106]. The conditioning medium of human MSC culture has a paracrine anti-apoptotic impact on hypoxia-induced death of rat lung alveolar cells. One of MSCs’ anti-apoptotic capabilities is the production of HGF and keratinocyte growth factor, which inhibit pro-apoptotic signals induced by reactive oxygen species (ROS) and hypoxia-inducible factor-1 alpha (HIF-1) [107].

MSCs simply release biologically active proteins and EVs that activate endogenous lung stem/progenitor cells, causing them to proliferate and differentiate, impede apoptosis, reduce inflammation, reestablish capillary barrier activity, and decrease fibrosis. They can treat both acute and chronic lung injuries since they function similarly to parental MSCs [108]. As a result, MSC-derived secretomes have anti-apoptotic properties, resulting in improved tissue healing and regeneration [109].

Multiple recent studies have discovered that the secretome of MSCs can control cell death, known as apoptosis, in both normal and diseased conditions. MSC-derived conditioned medium (MSC-CM) therapy is a promising approach in regenerative medicine and cell-based therapies [110]. It involves utilizing the secreted factors and soluble molecules present in the conditioned medium, which is the culture medium in which MSCs have been cultured. When MSCs are cultured in vitro, they release a variety of bioactive molecules such as growth factors, cytokines, chemokines, and EVs into the surrounding medium [111]. These secreted factors have been found to possess therapeutic properties, including anti-inflammatory, immunomodulatory, angiogenic, anti-fibrotic, and tissue repair-promoting effects. The application of MSC-CM therapy has shown promising results in preclinical and early clinical studies for various conditions, including but not limited to wound healing, tissue regeneration, neurological disorders, and immunomodulation [112,113,114].

The application of MSC-CM therapy demonstrated a decrease in the presence of pro-apoptotic markers, such as cleaved caspase-3 and Bax, within the main functional cells and concurrently increased the synthesis of the anti-apoptotic protein B-cell lymphoma/leukemia-2 (Bcl-2) [115,116]. This intervention prevented the elimination of these cells during instances of chronic inflammation. Administering MSC-CM through injection facilitated the regeneration of the liver and enhanced the survival rate in mice involved in the research. This effect was achieved by reducing the infiltration of inflammatory cells in the inflamed liver, minimizing apoptosis, and promoting the proliferation of damaged hepatocytes [117,118,119].

The inhibition of cell death and enhancement of liver cell regeneration caused by MSC-CM primarily occurred through the activation of signaling pathways such as HGF, fibrinogen-like protein 1, IDO-1/KYN, and IL-6/gp130. These pathways, along with other beneficial components derived from MSCs, exerted trophic and immunomodulatory effects, ultimately protecting the liver. Similarly, MSC-exosomes contributed to a decrease in negative immune responses and hepatocyte death in acute liver failure (ALF). Additionally, they hindered the production of TGF-β by hepatic stellate cells (HSCs), resulting in a reduction of liver fibrosis, similar to the effects observed with MSC-CM [120,121,122].

## 5. MSC-EVs as Diagnostics

Table 3 provides an overview of the diagnostic applications of EVs derived from MSCs in various neurological disorders. It emphasizes the potential uses of MSC-EVs as diagnostic tools and highlights specific examples supporting their application. In AD, MSC-EVs present promising opportunities for diagnostics and biomarker identification [123,124]. These EVs carry specific proteins and microRNAs that can be analyzed to identify early biomarkers of AD. Exosome-derived tau and amyloid-beta levels in CSF can indicate AD progression, while the detection of Aβ42 in CSF or blood aids in early diagnosis. Additionally, MSC-EVs can be utilized to track disease progression, monitor treatment response, and assess the effectiveness of therapeutic interventions in AD [123,124,125,126,127].

In PD, MSC-EVs also serve as potential diagnostics. Levels of α-synuclein and DJ-1 in MSC-EVs from CSF may indicate early signs of PD, and monitoring miR-34a levels in EVs can help track disease progression. Moreover, LRRK2-enriched EVs and miR-124-enriched EVs hold promise as potential biomarkers for PD detection [128,129,130].

In MS, quantification of myelin EVs in CSF shows promise as a potential biomarker for disease activity and progression. These myelin-derived EVs offer valuable insights into the disease’s status and advancement [131]. For ALS, MSC-EVs could aid in early diagnosis and prediction of disease outcomes by analyzing ALS-related proteins, including SOD1, TDP-43, pTDP-43, and FUS in these EVs [132]. Likewise, in HD, elevated total Huntingtin levels in EVs from plasma of HD groups compared to controls suggest the potential use of Huntingtin protein levels in EVs as a diagnostic biomarker for the disease [133].

These data highlight the immense diagnostic potential of MSC-EVs in neurological disorders, showcasing their capability to provide valuable information for early detection, disease monitoring, and treatment assessment.

**Table 3 biomolecules-13-01250-t003:** The diagnostic applications of MSC-EVs in neurological disorders.

Neurological Disorder	Diagnostic Uses and Applications	Examples	Reference
Alzheimer’s Disease	Biomarker Identification: EVs derived from MSCs carry specific proteins, microRNAs, and other molecules that can be analyzed to identify early biomarkers of Alzheimer’s disease.	Exosome-derived tau and amyloid-beta levels in CSF can indicate Alzheimer’s progression	[123,124]
	Detection of Aβ42 in CSF or blood to aid in early Alzheimer’s diagnosis.	Aβ42.	[127]
	Tracking disease progression, monitoring treatment response, and early detection.	Downregulation of miR-212 and miR-132-enriched EVs in AD	[126]
	Assessing the effectiveness of therapeutic interventions.	EVs containing Tau protein	[125]
Parkinson’s Disease	Biomarker Detection: EVs released by MSCs carry specific proteins and microRNAs that can serve as biomarkers to detect early signs of Parkinson’s disease.	α-synuclein and DJ-1 levels in MSC-EVs from CSF as a potential biomarker for Parkinson’s	[130]
	Monitoring disease progression by assessing miR-34a levels in EVs.	miR-34a over-expression in PD	[129]
	Biomarkers to detect PD.	miR-124-enriched EVs in PD	[128]
Multiple Sclerosis	Quantification of myelin EVs in CSF as a potential biomarker for disease activity and progression.	Myelin-derived EVs	[131]
Amyotrophic Lateral Sclerosis	Early diagnosis and predicting disease outcome.	ALS-related proteins, including SOD1, TDP-43, pTDP-43, and FUS	[132]
Huntington’s Disease	Elevated total Huntingtin levels in EVs from plasma of HD groups compared to controls.	Huntingtin protein	[133]

## 6. MSC-EVs in Alzheimer’s Disease

AD is a kind of dementia that typically affects the elderly. AD is a progressive neurological illness and the most frequent cause of dementia [134]. Damage is pervasive in AD, as many neurons cease to function, lose connections with other neurons, and die, affecting activities critical to neurons and their networks, including communication, metabolism, and repair. Initially, AD results in the deterioration of neurons and their synaptic connections in the brain regions associated with memory. As the disease progresses, it also impacts the regions of the cerebral cortex responsible for language, reasoning, and social interactions [135].

This neurodegenerative condition gradually and irreversibly impairs brain functioning (remembering, reasoning, and thinking), thought content, personality, and behavior [136]. The prevailing scientific explanations for the pathological features of AD involve the following processes: accumulation of Aβ outside cells, the creation of neurofibrillary tangles (NFT) caused by the buildup of hyperphosphorylated tau inside cells, and persistent neuroinflammation. These factors are considered to be the primary contributors to the development and progression of AD [137,138]. Modulation of the aberrant gene expression in AD can effectively improve the cognitive response in animal models of AD [139]. Neurotic communication abnormalities and the loss of individual neurons are caused by aberrant protein accumulations outside and inside nerve cells [138]. The first phases of AD pathogenesis are assumed to be caused by the deposition of Aβ, the primary component of amyloid plaque, within neurons [140]. These findings underline the importance of exosomes in the progression of AD through the spread of amyloid plaques [141]. In addition, the content of exosomes can modulate the gene expression of AD-associated genes.

Several research investigations have explored the use of exosomes as a potential biomarker for the early detection of AD (Figure 4). Additionally, they have been investigated as a means to transport therapeutic agents, such as small chemical molecule drugs, miRNA, and siRNA [142,143]. For instance, Saman et al. employed tau-containing exosomes generated from CSF for the initial diagnosis of AD [144]. Furthermore, since CSF-derived exosomes contain both p-tau and Aβ, the discovery of both possible biomarkers in exosomes may indeed raise the value of the currently employed marker for the initial detection of AD [145,146]. In this regard, using a combination of markers to diagnose AD enhanced specificity and sensitivity by 86% [147].

The AD sufferers’ CSF-derived exosomes had greater levels of miR-598 and miR-9-5p than their healthy counterparts [148]. Plasma, on the other hand, has long been evaluated to distinguish markers; hence, plasma seems to be a more convenient and accessible option [149]. Furthermore, the biological components of exosomes have demonstrated great accuracy in the initial detection of AD [150]. When comparing plasma exosomal protein expression among AD cases and healthy controls, researchers discovered that patients with AD had higher levels of neuron-derived proteins such as Aβ and tau [146].

Several investigations have confirmed the neuroprotective properties of neuron-derived exosomes. Exosomes generated from glia, for instance, protect neurons from oxidative stress [151]. According to the findings of a study into the processes involved with exosomes in Aβ clearance, neuron-derived exosomes injected into the brains of AD transgenic mice aided in Aβ peptide elimination. Because of conformational changes, the prion receptor on the exosomal surface can bind to amyloid plaques and convert them to harmless forms. Exosomes also hasten the absorption of Aβ extracellular plaque by microglia.

Exosomes released by MSCs obtained from connective tissues such as bone marrow and adipose tissue not only effectively traverse the blood–brain barrier (BBB) but also successfully degrade both intracellular and extracellular Aβ peptides in the brain. This degradation is attributed to the presence of neprilysin enzymes [152,153]. MSC-EVs directly interact with Aβ through their lipid membranes, promoting the clearance of Aβ plaques by microglia. Additionally, MSC-EVs transport neprilysin, an enzyme that breaks down Aβ, thereby indirectly reducing the accumulation of Aβ inside cells. In vivo studies investigating the therapeutic effects of MSC-EVs in animal models of AD have predominantly employed long-term treatment regimens lasting for weeks or months. The administration of EVs in these studies has been done either systemically (via the bloodstream) [154] or intracerebroventricular [155,156], demonstrating either partial recovery [154,155] or a protective function in diminishing the burden of Aβ plaque and the number of dystrophic neurites [156].

The injection of MSCs or MSC-EVs into hippocampal neurons enhances their resilience against the synaptic degradation caused by Aβ and the harmful effects of oxidative stress [157]. The results proposed several potential mechanisms to explain this phenomenon, including the decreased presence of extracellular Aβ due to the heightened endocytic capability of MSCs, the secretion of EVs containing antioxidant enzymes like catalase, and the paracrine activity resulting from the eventual release of trophic factors and anti-inflammatory cytokines such as VEGF, IL-6, and IL-10. Although most studies on AD treatment have primarily employed MSC-EVs, recent investigations suggest that different sources of stem cells possess therapeutic potential in combating AD-related cognitive disorders through distinct processes, such as reducing the extracellular and intracellular deposition of Aβ oligomers [157].

In a recent study, novel insights into the therapeutic mechanism of MSCs in AD were presented. They demonstrated that MSC-EVs derived from bone marrow transported miR-29c-3p to neurons, inhibiting the expression of beta-secretase (BACE1) and activating the Wnt/β-catenin pathway, ultimately leading to a beneficial effect in AD [158].

In their study, Wang et al. investigated the impact of MSC-EVs on neuronal impairments in hippocampal neurons stimulated by Aβ, as well as in AD cell lines (SHSY5Y) and AD animal models (APPswe/PS1dE9 mice) [159]. The results demonstrated that the administration of MSC-EVs effectively improved cognitive deficits, reduced the accumulation of Aβ in the hippocampus, and prevented neuronal loss in AD mice. Furthermore, they identified the involvement of the nuclear factor-erythroid factor 2-related factor 2 (Nrf2) signaling pathway in mediating the effects of MSC-EVs in both cell lines and animal models. These findings highlight the potential of MSC-EVs as promising nanotherapeutics for restoring the structure and function of hippocampal neurons in APP/PS1 mice. Based on the aforementioned findings, MSC-EVs demonstrate favorable effects in the context of AD [159].

Although MSC-EVs hold promise for treating AD, their full potential is hindered by several challenges. These issues include inadequate targeting efficiency, inconsistent treatment results, and limited production yield [160]. To overcome these limitations and enhance the effectiveness of MSC-EVs as AD treatments, it is essential to functionalize and engineer the EVs structures. Various methods can be employed for this purpose, such as preconditioning the parental cells to improve the natural treatment’s effectiveness, incorporating therapeutic cargo or drug loading into MSC-EVs, modifying the surface of MSC-EVs to enhance targeting capabilities, and utilizing artificial MSC-EVs to scale up production [161].

MSC-EV-based AD treatments were accomplished using different types of mesenchymal stem cells: bone marrow-derived MSCs (mBMSCs), adipose-derived MSCs (mADSCs), and human umbilical cord-derived MSCs (hUCMSCs). To enhance the therapeutic efficacy of MSC-EVs, they are modified with specific peptides like the rabies virus glycoprotein (RVG) peptide or loaded with miRNAs like miR-29 and miR-22 [161]. MSC-EVs were administered through various routes, including intravenous (IV), intracranial injection (IN), and dorsal hippocampus injection. IV administration of RVG-modified MSC-EVs has shown promise in improving learning and memory by reducing Aβ deposition and astrocyte activation while promoting the production of anti-inflammatory factors. Similarly, dorsal hippocampus injection of miR-29-loaded MSC-EVs reduces BACE1 expression and activates PKA/CREB, leading to improved cognitive function [162]. Preconditioning MSCs in an AD environment before administration is another promising strategy. Hypoxic preconditioning of mADSCs shifts microglial M1/M2 polarization, reduces inflammatory factors, and upregulates TREM2 expression, thereby improving cognitive function [163].

Although most studies have used MSC-EVs for AD treatment, it is worth noting that various stem cell sources have demonstrated therapeutic potential in alleviating AD-associated cognitive deficits via multiple mechanisms, such as reducing extracellular and intracellular oligomer deposition (Table 4).

## 7. MSC-EVs in Parkinson’s Diseases

PD is the second most prevalent neurodegenerative disease worldwide, originally discovered by James Parkinson in 1817. While the exact causes of PD are still unknown, both genetic and environmental factors contribute to its development [171,172]. Degeneration of dopaminergic neurons and impairment of dopamine production in multiple dopaminergic networks characterize the disease. The formation of Lewy bodies in the neurological system, which are protein clumps formed of α-synuclein (α-syn), is linked to the death of dopaminergic neurons and the disturbance of their normal functioning. The nigrostriatal pathway, which includes the substantia nigra pars compacta and the striatum, is the most damaged [173]. There is currently no cure for PD, but researchers are exploring various treatment options, including the use of MSCs and their EVs [174,175,176,177].

Exosomes can transport enzymatically active proteins like phosphatase and tensin homolog (PTEN) and biologically active lipids, such as prostaglandins, to specific target cells [178,179]. Within exosomes, there are various proteins referred to as “exosome markers,” which are primarily involved in their formation. These exosomes also carry transmembrane molecules that assist in the immunoselection process, enabling the identification of exosomes with a specific biological origin and enhancing their sensitivity as biomarkers. In the context of NDs like PD, exosomes have been found to contain misfolded proteins such as α-syn [180,181].

The presence of genetic material, such as miRNAs, is one of the largest common contents of exosomes [74]. Disorders like PD exhibit considerable disruption in gene expression, especially at the miRNA level [182,183]. Exosomes derived from MSCs can transfer miRNAs to neuronal cells. Notably, exosomes rich in miR-133b have been found to promote neurite outgrowth, which is advantageous for PD since this particular miRNA is generally suppressed in PD cases [166]. However, it should be noted that MSC-derived exosomes also contain miR-143 and miR-21, which are known to play significant roles in regulating immune responses and contributing to neuronal loss associated with chronic inflammation [184].

The α-syn can be released either directly into the extracellular space or enclosed within exosomes [185]. Furthermore, α-syn has been observed to engage with synaptic vesicles, leading to an augmentation of neurotransmitter release and assisting in the assembly of SNARE proteins. Typically, synaptic vesicles that contain α-syn are sorted into early endosomes either through Golgi or clathrin-mediated endocytosis [186,187]. The endosomes that contain α-syn progress and develop into multivesicular bodies (MVBs). These MVBs eventually merge with the cell membrane and release their contents as exosomes, aided by VPS4 and small ubiquitin-like modifier (SUMO) proteins. Another possibility is that the α-syn-containing endosomes are directed towards recycling endosomes, where they are released from the cell as secretory granules through a process dependent on Rab11a [188,189]. While engaged in these processes, the level of calcium in the cytoplasm governs the discharge of α-syn from viable cells. Although the quantity of α-syn found in exosomes is limited, recent studies propose that exosomes create a favorable setting for the aggregation of α-syn, potentially contributing to the propagation of PD (Figure 5). The toxic variant of α-syn, known for its ability to trigger neuronal cell demise, is typically recognized as oligomeric α-syn [190,191].

Currently, the identification of PD primarily relies on the observation of visible motor symptoms during clinical examination. Unfortunately, there are no reliable diagnostic methods available for detecting PD in its early stages. Developing a technique for early detection would be a significant advancement in the field. Previous studies have indicated that certain components of EVs, such as exosomes obtained from the blood or CSF of PD patients, can serve as effective biomarkers for the disease [192,193,194]. It is crucial to comprehend the complexity of exosomes derived from MSCs and how their miRNA contents interact with the cellular and molecular pathways associated with PD.

MSC-EVs have been proposed as a promising therapeutic tool for PD, as they can act as a vehicle for the delivery of therapeutic molecules, such as miRNAs, to the brain [174]. MSC-EVs could be modified using molecular engineering techniques to carry protein and RNA cargoes, making them a promising therapeutic option for PD [175]. MSC-derived secretome treatment has shown encouraging results in experimental models of PD [176].

MSCs and the EVs they produce have been suggested as a viable treatment approach for various neurodegenerative conditions such as PD. This is because they possess the capability to support the survival of dopaminergic neurons, encourage the formation of new neurons, decrease neuroinflammation, and improve overall functional recuperation in animal models [177]. In a pilot investigation, individuals diagnosed with progressive supranuclear palsy (a rare and serious type of parkinsonism) received mesenchymal stem cells derived from bone marrow. These cells were administered through the cerebral arteries. The results showed that all of the treated patients survived for a year following the infusion of the cells, except for one patient who passed away nine months later due to reasons unrelated to the delivery of cells or the progression of their illness [195].

One of the major obstacles to PD treatment is access to the damaged cells. However, recent bioengineering research has led to the production of genetically modified cells with improved therapeutic efficiency. Improved adhesion, migration, and survival are new methods for not only preserving but also increasing the biological properties and therapeutic potential of MSCs [196]. MSCs that have been genetically designed to produce specific neurotrophic factors, such as brain-derived neurotrophic factors, or MSCs that have been modified to boost their survival and ability to migrate toward the lesion location. Concerning PD, various studies have utilized engineered MSCs expressing vascular endothelial growth factor, tyrosine hydroxylase, or modified to enhance the production of cerebral dopamine neurotrophic factor or glial cell-derived neurotrophic factor. These experiments have shown promising outcomes in preclinical rodent models [197]. Furthermore, genetically engineered EVs showed promising results in PD animal models. When subjected to catalase-loaded EVs in a cell culture environment, macrophages that were activated with lipopolysaccharide (LPS) and tumor necrosis factor (TNF) demonstrated decreased levels of reactive oxygen species (ROS). In a mouse model of PD using 6-OHDA, administering catalase-loaded EVs resulted in reduced activation of microglia compared to the application of free catalase. These findings suggest that delivering catalase through EVs has the potential to effectively decrease oxidative stress and neuroinflammation in PD [198]. Interestingly, the introduction of dopamine-loaded EVs led to a remarkable 15-fold increase in dopamine distribution within the brain. This heightened distribution not only resulted in improved therapeutic effectiveness but also significantly reduced systemic toxicity compared to the administration of free dopamine. These findings suggest that utilizing EV-based drug delivery holds promising potential as a viable and effective treatment option for PD [199].

## 8. MSC-EVs in Multiple Sclerosis

MS is inflammatory demyelination of the central nervous system [200] (Figure 6). In addition to inflammation and demyelination in the spinal cord and brain, other pathological biomarkers of MS include BBB disruption, reactive gliosis, oligodendrocyte loss, and neuron and axonal degeneration [201]. According to the National MS Society, the four major types of MS are primary progressive MS (PPMS), secondary progressive MS (SPMS), relapsing–remitting MS (RRMS), and clinically isolated syndrome (CIS) (NMSS) [202].

Autoimmune attacks are the most significant reason for axon demyelination in illnesses like MS. Ideally, tissue restoration via stem cell transplantation may lead not only to axon reconstruction by replacing lost and destroyed cells but also to anti-inflammatory and paracrine neuroprotective effects, potentially preventing progressive neural and axonal degeneration [203,204].

The exosomes derived from periodontal ligament stem cells and present in the conditioned medium demonstrated anti-inflammatory and suppressive effects in mice models of MS called experimental autoimmune encephalomyelitis (EAE). The study indicated that the exosomes promoted significant remyelination in the spinal cord and reversed the progression of MS by increasing the levels of anti-inflammatory cytokines, specifically IL-10. Moreover, the impaired activation of T cells, which is a key factor in regulating the balance between T helper (Th)1 and Th2 cells, was identified as one of the pathological features of MS [205].

The EVs produced by placental-derived MSCs (PMSCs) could achieve therapeutic effects similar to individual EAE therapy if administered in high doses. Also, according to VEGF proteomic analyses, the HGF was found in EVs produced from PMSC. PMSCs regulate the immune system by triggering regulatory T cells (Tregs) with large amounts of these substances that they release. This discovery indicated that PMSC-EV can stimulate myelin regeneration and elicit immunomodulatory effects comparable to PMSC therapy in the EAE mice model [170]. Likewise, the administration of MSC-EVs obtained from adipose tissue of humans through intravenous treatment improves the condition of animals with EAE. This is achieved by inhibiting the infiltration of immune cells, modulating their activation, and reducing the secretion of inflammatory cytokines [206].

Induction and maintenance of immunological tolerance is a major goal in the treatment of autoimmune conditions. Maintenance and progression of regulatory molecules like TGF-β, programmed death ligand-1 (PD-L1), and galectin-1 via biological interventions in the host immune system are one of the most current approaches for peripheral tolerance [206,207]. The ability of MSC-MV to enhance environmental resilience in splenic mononuclear cells (MNCs) from mice with EAE was investigated. The MVs derived from MSCs trigger apoptotic signaling in self-reactive lymphocytes, prompting them to release IL-10 and TGF-β. Additionally, they upregulate the expression of regulatory molecules like TGF-β and PD-L1 on the surface of MVs, which promotes the differentiation of regulatory T cells (Tregs). This ultimately contributes to the development of peripheral immune tolerance [168].

A different developing approach to induce immunological tolerance in individuals with MS involves directing microglia to adopt the M2 phenotype [208]. Microglia are the CNS’s resident macrophages that are rapidly activated by microenvironments (like infection, ischemic injury, and pro-inflammatory cytokines such as TNF-α and IL-1β) to make a differentiation either into the M1 phenotype, which causes CNS damage and generates pro-inflammatory cytokines, or the M2 phenotype, which fosters tissue regeneration by producing anti-inflammatory cytokines [209].

In the early stages of MS, an imbalance of M1/M2 macrophages and a shift toward pro-inflammatory M1 phenotypes was considered to be one of the major drivers of tissue injury in the CNS. As a result, it is thought that prompting microglia to polarize toward the M2 phenotypes might improve MS patients’ neurological symptoms [208]. In this regard, Li et al. investigated the impact of BMSC paracrine pathways, namely exosome mediation, on microglial polarization and motor functional improvements in an EAE mouse model [169]. They found that exosomes derived from BMSC can decrease demyelination and inflammation of the CNS while improving neural behavioral ratings in the EAE animal model via shifting the polarity of microglia toward an M2 phenotype. Furthermore, MSC exosome therapy decreased M1-associated TNF and IL-12 levels while increasing M2-associated cytokines (IL-10 and TGF-β) [169,210].

EVs derived from MSCs of adipose tissue promote recovery from demyelination in an animal model for progressive MS, and lab animals induced recovery from demyelination and curation of brain atrophy [104]. MSC- EVs have the potential to exert positive effects by transporting crucial molecules, including DNA, enzymes, proteins, mRNA, ncRNAs, and different ligands, to the intended recipient cells [211]. The field of molecular engineering has made alterations to EVs by incorporating myelin antigens. This modification transforms EVs into platforms capable of presenting antigens, thereby enabling the restoration of antigen-specific peripheral immune tolerance in autoreactive T cells. This innovative technique can be regarded as a groundbreaking “EV-based vaccine” that holds significant potential in the treatment of MS by reinstating immune tolerance. Antigen-presenting EVs could decrease harmful immune reactions while preserving the integrity of the remaining immune system, thereby minimizing the likelihood of adverse outcomes [211].

Exosomes can carry medications to MS patients because of their capacity to cross the BBB. Diverse functional elements on the surfaces of exosomes, such as aptamers and antibodies, dramatically improve the exosomes’ specificity [212]. Based on these findings, it’s safe to say that MSC-EVs will represent the future of the MS therapy approaches for various reasons, such as their safety and capacity to cross the BBB. 

The therapeutic effects of genetically engineered MSCs in different MS models were investigated. Female mice were treated with Mouse MSCs expressing the Mouse IFN-β gene. Intravenous administration of engineered MSCs resulted in increased Tregs and IL-10 production while reducing inflammatory cell infiltration, suggesting potential therapeutic benefits for MS [213]. Furthermore, treatment with human BM-MSCs engineered to express PSGL-1, FUT-7, and IL-10 resulted in an increase in clinical score and myelination, coupled with reduced inflammatory infiltration. However, the observed impact of the engineered MSCs on MS pathogenesis appears complex, necessitating further research for a comprehensive understanding [214].

## 9. MSC-EVs in Amyotrophic Lateral Sclerosis

ALS is a fatal neurodegenerative condition that typically develops in adulthood and was first identified in the 1870s. While around 5% to 10% of people with ALS have a family history of the disease (known as familial ALS or fALS), the remaining 90% to 95% of cases (referred to as sporadic ALS or sALS) do not seem to have a clear genetic connection [215]. The patient’s condition worsens and becomes life-threatening within a period of 2 to 5 years after the disease begins. Both sporadic sALS and familial fALS have a shared characteristic of experiencing a targeted loss of upper motoneurons in the primary motor cortex, as well as lower motoneurons in the brainstem and spinal cord. However, the disease does not affect specific motoneurons that control pelvic muscles and eye movements. The exact reason for this differing vulnerability of motoneurons is presently unknown [216].

Neurodegeneration is characterized by a complex underlying mechanism involving various pathways. One such pathway involves the increased entry of calcium ions (Ca^2+^) into motoneurons, which is triggered by elevated levels of the neurotransmitter glutamate in the synaptic cleft (known as glutamate excitotoxicity) caused by dysfunction in the uptake process by astrocytes. Due to problems with mitochondrial function, the concentration of Ca^2+^ remains elevated within the cytoplasm, leading to the activation of enzyme pathways dependent on Ca^2+^ and contributing to oxidative stress, potentially leading to dementia.

Exosomes and MVs have received increased attention as boosters and suppressors of disease processes due to their ability to transmit biological information across large distances. Exosomes derived from primary neurons or neuroblastoma have been shown to improve the course of AD in a mouse model by sequestering intracerebral substances. After oxidative stress, the neuroprotective effect of exosomes produced from adipose-derived stromal cells (ASC) was also established in primary murine hippocampus neurons and human neuroblastoma cells. Exosomes derived from BM-MSC have also been used to aid recovery and neuroregeneration following strokes and traumatic brain injuries. Exosomes derived from ASC have recently been shown to protect neurons in an in vitro model of ALS [16,217,218].

The NSC-34 cell line, which mimics motoneurons affected by ALS, was genetically modified with different SOD1 point mutations to replicate the characteristics of the disease. In the research, H2O2 was utilized as a harmful stimulus. The lifespan of ALS motoneurons was enhanced by exosomes, which inhibited the apoptotic pathway. This suggests that exosomes have the potential to be utilized as a therapeutic approach for ALS. Another investigation demonstrated that exosomes derived from ASCs could potentially treat ALS by reducing the presence of mutant SOD1 and enhancing the functioning of mitochondrial proteins involved in aggregation [219]. Although the researchers still have a long path ahead of them, the recent discovery of EVs gives patients with ALS new hope and should stimulate more research in this approach.

Using EVs derived from MSCs in ALS treatment has several advantages over using MSCs themselves. EVs can cross the blood–brain barrier, which is a significant challenge for MSCs [220]. EVs can be stored and transported more easily than MSCs [221]. EVs can be produced in large quantities and standardized more easily than MSCs, and EVs have a lower risk of immune rejection than MSCs [221]. In addition, EVs can deliver therapeutic molecules to target cells, such as microRNAs, which can regulate gene expression and promote neuroprotection [222].

## 10. MSC-EVs in Huntington’s Disease

The progressive loss of brain cells in the putamen, caudate, and cerebral cortex caused by HD, a hereditary neurodegenerative condition, results in physical, mental, and emotional problems. The IT-15 gene has dominant mutations that encourage the development of poly-glutamine (polyQ) repeat sequences in Huntingtin proteins, specifically by boosting the number of CAG repeats inside a polyQ repeat gene sequence. Huntingtin interacts with about 100 other proteins, which suggests that it participates in a variety of biological activities [223]. Polyglutamine Huntingtin protein is indeed transported to other cells by the exosome in HD. As a result, exosomes are crucial in the advancement of HD pathogenesis.

Exosomes have been explored as potential treatments for HD [224,225]. Lee and colleagues conducted research in this area and observed that exosomes derived from adipose-derived mesenchymal stem cells (ADMSC) can regulate harmful characteristics in HD cell models. These exosomes were found to reduce the presence of mHtt intracellular aggregates and increase the expression levels of PGC-1 and phospho-CREB (cAMP response element-binding) [226]. Additionally, the same research group investigated the delivery of miR-124 through exosomes to the striatum of R6/2 HD transgenic mice. Despite observing a decrease in the intracellular expression of the miR-124 targeted gene, REST, the effects on the mice’s behavior were minimal [227].

Studies have shown that MSC-EVs have particular effects on HD. In vitro analysis has revealed that MSC-EVs can constrain motor function and striatal atrophy in a rat model of HD [228]. In their study, Ebrahimi and colleagues showed that the release of GDNF and vascular endothelial growth factor (VEGF) from MSCs had a positive effect on motor coordination and muscle functions in animal models of HD [229].

A scalable and dependable technique for loading therapeutic RNA into extracellular vesicles (EVs) has been devised. This method involved incorporating a hydrophobically modified siRNA, designed to target Huntington RNA, into the EVs without causing any adverse effects on their size or structural integrity. The effectiveness of this approach was demonstrated by efficiently silencing Huntington mRNA both in vitro using mouse primary cortical neurons and in vivo after administration into the mouse striatum [230].

## 11. Challenge and Future Perspectives

Caution should be exercised when interpreting studies comparing the effectiveness of MSC-EVs and their cellular counterparts due to limitations in current methods of EV quantification [231]. Furthermore, there is insufficient evidence regarding the optimal therapeutic dosage of EVs and their long-term effects [232]. Some types of tissue damage may not require multiple administrations of MSC-EVs, while others may necessitate repeated treatments, which can increase the burden and expenses for the patient. Addressing the challenge of heterogeneity among MSC-EVs is crucial before their widespread clinical application. Different types of MSC-EVs have varying levels of treatment efficacy, underscoring the need for further research to identify subgroups with the greatest therapeutic potential. Improved methodologies and research in genomics, proteomics, and other fields are necessary to accurately classify and differentiate EV subtypes. Additionally, it is important to enhance the reproducibility of large-scale EV production with high purity and specific therapeutic activity. This calls for the development of robust in vitro quality control systems tailored to the specific requirements of EV-based treatments [232].

The MSC-EVs have emerged as a promising area of research in regenerative medicine and therapeutic applications [233,234]. These tiny membrane-bound vesicles carry bioactive molecules and can influence the behavior of target cells and tissues. Despite their potential, MSC-EVs face challenges in standardization, scalability, payload loading, storage, and targeting [235,236]. However, the future prospects for MSC-EVs are exciting, with potential applications in treating various diseases, tissue regeneration, immunomodulation, personalized medicine, drug delivery, and as biomarkers for diagnostics. The study of MSC-EVs has also advanced nanomedicine and drug delivery systems. Ongoing research and clinical trials may lead to the approval of MSC-EV-based therapies and contribute to significant advancements in the field of regenerative medicine and therapeutics.

## 12. Conclusions

The therapeutic potential of MSC-EVs has been demonstrated in various cases of tissue injury. In vivo studies have shown that MSC-EVs are as effective as the parent cells in promoting tissue regeneration. This finding is intriguing for the potential clinical use of MSC-EVs, suggesting that they could be a cost-effective alternative to MSC-based therapies. Current preclinical research suggests that MSC-EVs directly target macrophages and the injured tissue, indicating that their positive impact on tissue regeneration is achieved through the modulation of immune responses and direct interaction with the tissue. A deeper understanding of this combined activity of MSC-EVs can facilitate better targeting of specific tissues and enhance treatment efficacy. Importantly, uncovering the mechanism of action of MSC-EVs is crucial for determining their legal status and maximizing their therapeutic effects, which are vital considerations for their potential clinical applications.

## Figures and Tables

**Figure 1 biomolecules-13-01250-f001:**
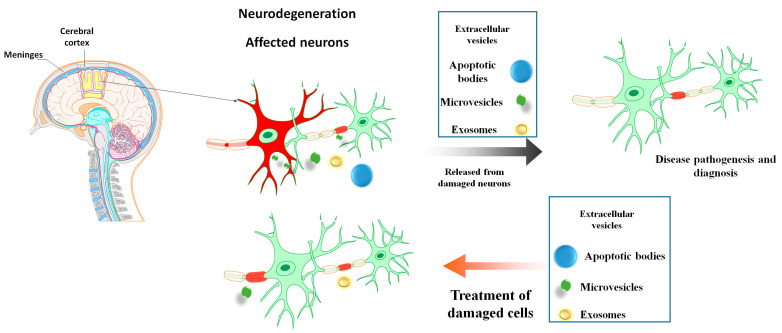
The various forms of extracellular vesicles. Exosomes, microvesicles, and apoptotic bodies play critical roles in the diagnosis and treatment of neurological disorders. Extracellular vesicles can easily pass across the blood–brain barrier and reach their destination in the injured cells. Extracellular vesicles released from damaged cells have a dual role in neurological disorders. On the one hand, they can contribute to the spread of pathogenesis, while on the other hand, they can serve as valuable biomarkers for these diseases.

**Figure 2 biomolecules-13-01250-f002:**
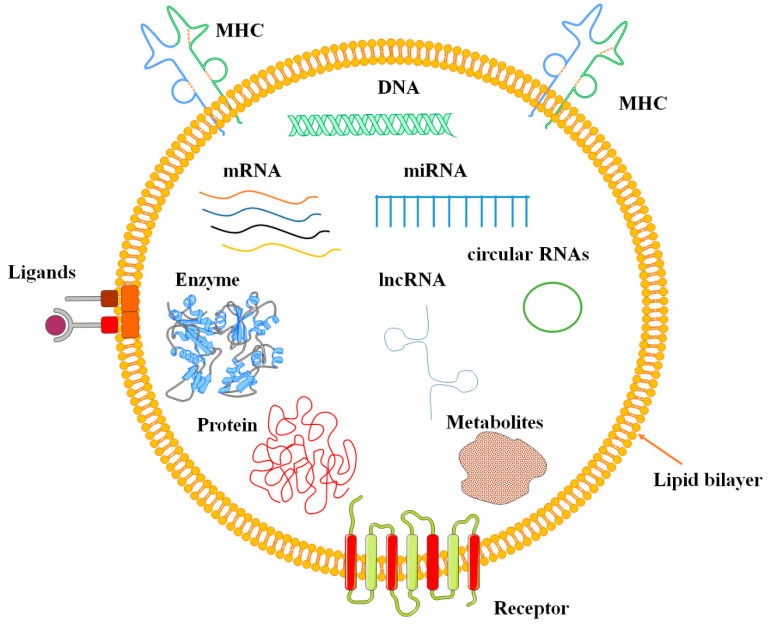
EVs characteristics and content. Cells release lipid-bound vesicles, known as EVs, into the extracellular environment. EVs can contain microRNAs, mRNAs, circular RNAs, long noncoding RNAs, proteins, lipids, and metabolites.

**Figure 3 biomolecules-13-01250-f003:**
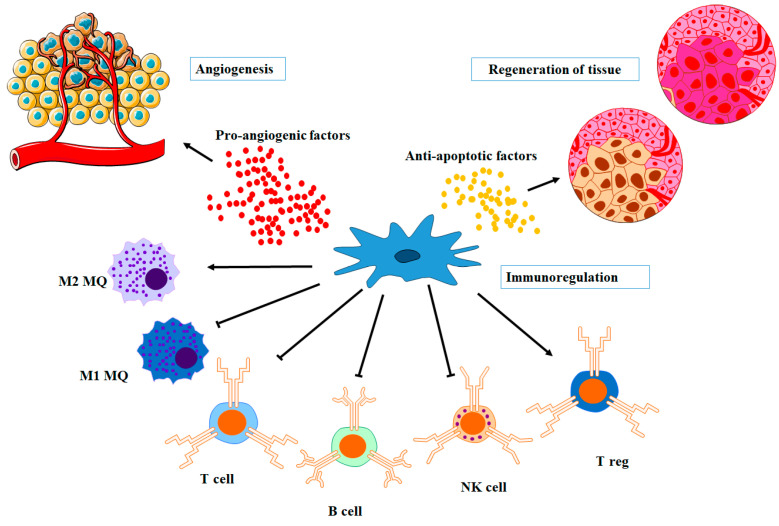
The components of MSC-EVs, as well as their major interactions. Angiogenesis, immunological modulation, and tissue regeneration are the three major modes of action of EVs.

**Figure 4 biomolecules-13-01250-f004:**
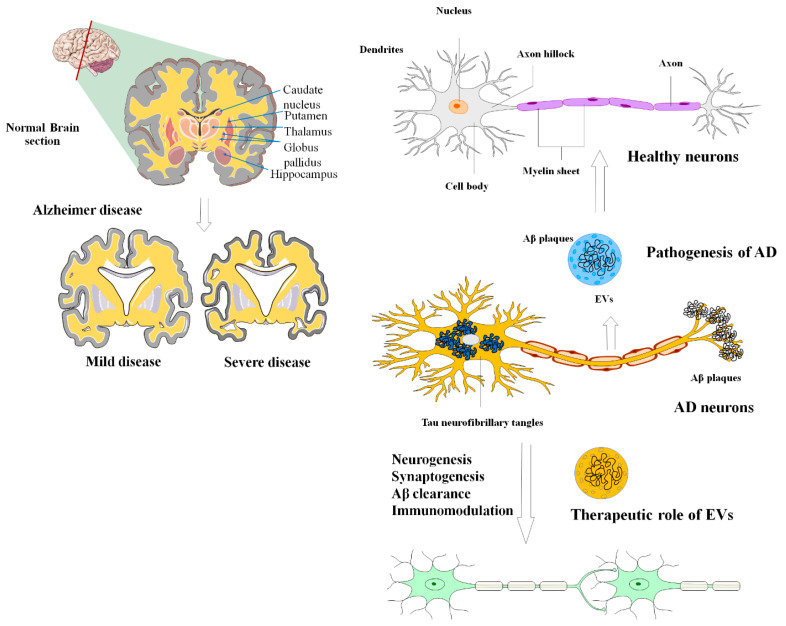
The applications and uses of MSCs-EVs in the diagnosis and treatment of AD. The most widely accepted scientific explanations for the pathogenic hallmarks of AD are extracellular aggregation of beta-amyloid peptide (A), formation of neurofibrillary tangles due to intracellular deposition of hyperphosphorylated tau, and persistent neuroinflammation. EVs can effectively alleviate these negative circumstances during the development of AD.

**Figure 5 biomolecules-13-01250-f005:**
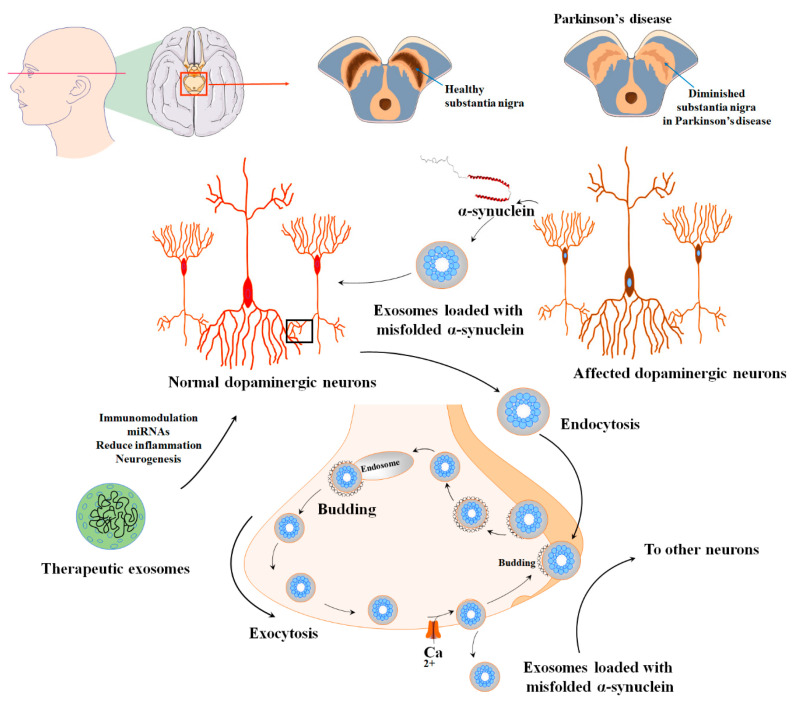
The various applications and uses of extracellular vesicles derived from mesenchymal stem cells in the diagnosis and treatment of PD. PD is characterized by the degeneration of dopaminergic neurons and impaired dopamine production within multiple dopaminergic networks. This condition is linked to the formation of Lewy bodies in the nervous system, which are aggregates of α-synuclein protein and result in the degeneration of dopaminergic neurons and disruption of their normal functioning. The nigrostriatal pathway, consisting of the substantia nigra pars compacta and the striatum, is particularly affected in PD. α-syn-containing synaptic vesicles are usually sorted into early endosomes through Golgi or clathrin-mediated endocytosis. However, extracellular vesicles show promise in effectively alleviating these detrimental conditions during the pathogenesis of PD.

**Figure 6 biomolecules-13-01250-f006:**
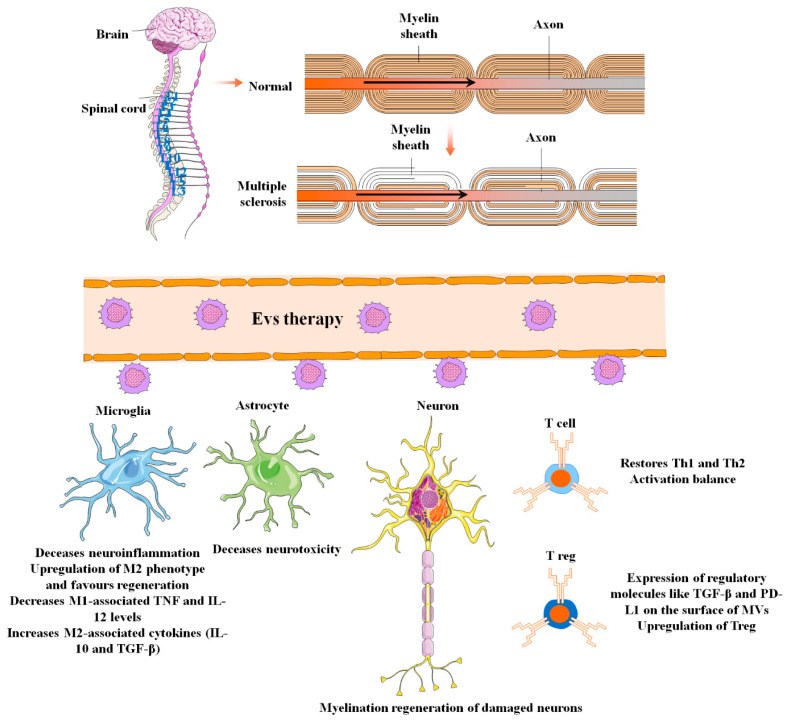
The applications and uses of mesenchymal stem cell-derived extracellular vesicles in the diagnosis and treatment of multiple sclerosis (MS). MS is central nervous system demyelination caused by inflammation. Other pathological biomarkers of MS, in addition to inflammation and demyelination in the spinal cord and brain, include BBB disruption, reactive gliosis, oligodendrocyte loss, and neuron and axonal degeneration. EVs can effectively relieve these adverse conditions during MS pathogenesis through several mechanisms of action.

**Table 1 biomolecules-13-01250-t001:** Types of vesicles. A concise overview of the sizes and origins of different vesicles, highlighting their unique characteristics and potential roles in cellular processes.

MSC-EV	Structure	Content	Origin	Applications
Exosomes	Small lipid bilayer vesicles (30–150 nm)	miRNAs, mRNAs, proteins, lipids, and signaling molecules	Luminal budding into MVBs; release by fusion of MVB with cell membrane	Therapeutic delivery, tissue regeneration, immunomodulation, and drug delivery
Microvesicles	Larger vesicles (100–1000 nm) formed by outward budding and shedding from the cell membrane	Proteins, lipids, mRNA, miRNA, and DNA	Outward budding of cell membrane	Tissue repair, wound healing, and immunomodulation
Apoptotic bodies	Large vesicles (50 nm–1 µm) released during apoptosis	Nucleic acids, histones, and fragmented organelles	Outward blebbing of apoptotic cell membrane	Immunomodulation, tissue regeneration, and biomarkers for cell death

**Table 4 biomolecules-13-01250-t004:** The use of extracellular vesicles produced from stem cells in neurodegenerative disorders.

Disease	Type of EVs and Origin	Outcomes	Ref
AD	MSCs/exosomes	Enhances neurogenesis, reduces Aβ, and the restoration of cognitive function.	[164,165]
PD	MSCs/exosome	Transferring of the miR-133b regulates neurite outgrowth.	[166]
		Improved neuronal function and oligodendrogenesis stimulation	[101]
		Reduction in α-syn aggregates	[167]
MS	MSCs/exosomes MSCs/EVs	Drive peripheral resistance, activate apoptotic signaling pathway in self-reactive lymphocytes, and stimulate regulatory T cell differentiation by - IL-10 and TGF-β secretion - expression of PD-L1 and TGF-β	[168]
	MSCs/exosomes	Reduce CNS inflammation and demyelination by performing the following: - Shifting microglial polarization toward an M2 phenotype	[169,170]

## Data Availability

All data are in the manuscript.

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
