# Peer review of "Mesenchymal Stem Cell-Derived Extracellular Vesicles: An Emerging Diagnostic and Therapeutic Biomolecules for Neurodegenerative Disabilities"

_biomolecules, 2023, doi:10.3390/biom13081250_

Round 1
Reviewer 1 Report
In their paper entitled “Mesenchymal Stem Cell-Derived Extracellular Vesicles: An Emerging Diagnostic and Therapeutic Biomolecules for Neurodegenerative Disabilities”, the Authors discuss the therapeutic potential of EVs generated from MSC cells in the treatment of a number of neurodegenerative diseases, as well as the already existing data concerning their real efficacy.
The paper is of interest and suitable for Biomolecules. The state of art is well described, and many interesting and recent findings have been reported. Bibliography is up-to-date: 53 (out of 166) of the cited papers have been, indeed, published between 2019 and 2023. Moreover, the concepts discussed are also illustrated by explicative figures.
Main comments:
-The addition of a further table, reporting the miRNAs found in MSC-derived EVs could be of help for the readers;
Minor comments:
Line 112: “endoplasmic reticulum fragments” should be more correct;
Line 242: “including as…”: the word as should be deleted;
Lines 424-425: the sentence “Even though autoimmune attacks are the most significant reason for axon demyelination in illnesses like MS” lacks a principal sentence;
Lines 519-520: Are the Authors referring to AD (again) or to ALS in this sentence?
Author Response
REVIEWER 1
In their paper entitled “Mesenchymal Stem Cell-Derived Extracellular Vesicles: An Emerging Diagnostic and Therapeutic Biomolecules for Neurodegenerative Disabilities”, the Authors discuss the therapeutic potential of EVs generated from MSC cells in the treatment of a number of neurodegenerative diseases, as well as the already existing data concerning their real efficacy.
The paper is of interest and suitable for Biomolecules. The state of art is well described, and many interesting and recent findings have been reported. Bibliography is up-to-date: 53 (out of 166) of the cited papers have been, indeed, published between 2019 and 2023. Moreover, the concepts discussed are also illustrated by explicative figures.
Response to reviewer comments
The authors would like to thank the reviewer for the positive comments
Main comments:
-The addition of a further table, reporting the miRNAs found in MSC-derived EVs could be of help for the readers;
Response to reviewer comments
A table and several paragraphs were added to demonstrate the important role of MSC-derived EVs miRNAs
The following part was added to the text.
The miRNAs contained in MSC-derived EVs have been shown to play a vital role in the therapeutic function of MSC-derived EVs in a number of diseases. They have immense potential and practical value in treating several diseases. Table 2 provides a comprehensive overview of various miRNAs found in MSC-derived EVs, along with their roles and potential applications. These miRNAs demonstrate diverse functions and exhibit promising therapeutic potential in different areas of regenerative medicine and disease treatment. One notable miRNA is miR-22, which inhibits the inflammatory response and contributes to nerve function recovery. It has the potential to reduce the release of inflammatory factors and minimize infarct size, suggesting its application in conditions associated with inflammation and tissue damage. Another miRNA, miR-21, plays a crucial role in promoting cell survival and proliferation. Its potential applications lie in tissue regeneration and wound healing, indicating its therapeutic importance in the field of regenerative medicine [66,67].
MiR-let-7, a regulator of cell proliferation and differentiation, shows promise in tissue regeneration and angiogenesis. This miRNA holds potential for therapeutic interventions aimed at promoting tissue repair and vascularization. Furthermore, miR-29b-3p is involved in modulating extracellular matrix remodeling. Its potential applications encompass fibrosis treatment, tissue repair, and fracture healing, highlighting its importance in addressing conditions characterized by abnormal tissue remodeling.
MiR-126 demonstrates a role in promoting angiogenesis and neurogenesis. This miRNA holds promise for cardiovascular regeneration and potential therapeutic applications in neurological disorders. Additionally, miR-133 exhibits anti-inflammatory properties and has the potential to reverse liver injury. Its application may be targeted towards liver injury treatment. MiR-146a-5p and miR-155 are involved in modulating immune responses and inflammation. These miRNAs have the potential for application in autoimmune diseases and inflammatory modulation therapies.
MiR-210 induces vascularization and promotes cellular adaptation to hypoxia. It holds promise for the treatment of ischemic diseases, tissue repair, bone regeneration, and selective regeneration of ischemic heart tissues. MiR-223 regulates neuronal cell apoptosis and immune cell reactions, presenting potential applications in conditions such as ischemic kidney and myocardial infarction. Moreover, the miR-17-92 cluster is associated with increased neural plasticity and functional recovery, particularly in treating stroke.
These findings on miRNAs found in MSC-derived EVs provide valuable insights into their roles and potential applications in various therapeutic areas, paving the way for further research and development in regenerative medicine and disease treatment.
Table 2. The miRNAs found in MSC-derived EVs, their role and applications.
|
miRNA |
Role/Function in MSC-derived EVs |
Potential Applications |
Reference |
|
miR-22 |
inhibits the inflammatory response and nerve function recovery |
inhibit the release of inflammatory factors, reduction of infarct size |
[66,67] |
|
miR-21 |
Promotes cell survival and proliferation |
Tissue regeneration, wound healing |
[68] |
|
miR-let-7 |
Regulates cell proliferation and differentiation |
tissue regeneration, angiogenesis |
[69] |
|
miR-29b-3p |
Modulates extracellular matrix remodeling |
Fibrosis treatment, tissue repair and fracture healing |
[70] |
|
miR-126 |
Promotes angiogenesis and neurogenesis |
Cardiovascular regeneration, neurological disorders |
[71] |
|
miR-133 |
Reduced inflammation and reversed liver injury |
Liver injury |
[63] |
|
miR-146a-5p |
Modulates immune responses and inflammation |
Autoimmune diseases, inflammation modulation |
[72] |
|
miR-155 |
Modulates immune responses and inflammation |
Immunomodulation, inflammatory disease therapy |
[73] |
|
miR-210 |
Induce vascularization, Promotes cellular adaptation to hypoxia |
Ischemic diseases, tissue repair, bone regeneration, selective regeneration of ischemic heart |
[74,75] |
|
miR-223 |
Regulates neuronal cell apoptosis and immune cells reactions |
Ischemic kidney, myocardial infarction |
[76-78] |
|
miR-335 |
Promotes osteoblast differentiation |
bone fracture recovery |
[79] |
|
miR-486 |
Promotes cardiac regeneration and repair |
Heart disease treatment, cardiac tissue repair |
[80] |
|
miR-499 |
Inhibit endometrial cancer growth and metastasis |
anticancer |
[81] |
|
miR-17-92 cluster |
Increases neural plasticity an functional recovery |
In treating stroke |
[82] |
Minor comments:
Line 112: “endoplasmic reticulum fragments” should be more correct;
Response to reviewer comments
Corrected
Line 242: “including as…”: the word as should be deleted;
Response to reviewer comments
Corrected
Lines 424-425: the sentence “Even though autoimmune attacks are the most significant reason for axon demyelination in illnesses like MS” lacks a principal sentence;
Response to reviewer comments
Corrected as follows “Autoimmune attacks are the most significant reason for axon demyelination in illnesses like MS. Ideally,………………………….”
Lines 519-520: Are the Authors referring to AD (again) or to ALS in this sentence?
Response to reviewer comments
We thank the reviewer for this comment.
The sentence is deleted
“Additionally, the formation of misfolded proteins resulting from genetic mutations leads to the aggregation of these proteins in cells, exacerbating oxidative stress, impairing mitochondrial function, and potentially resulting in an accumulation of neurofilaments (NFs) and dysfunction in axonal transport. Furthermore, the release of inflammatory mediators and toxic substances by activated astrocytes and microglia further contributes to neurotoxicity [169].
”

Reviewer 2 Report
The manuscript entitled “Mesenchymal Stem Cell-Derived Extracellular Vesicles: An Emerging Diagnostic and Therapeutic Biomolecules for Neurodegenerative Disabilities” is a review article that has described different EVs and their potential therapeutic effects in neurodegenerative diseases. The manuscript has some major concerns. What is the novelty in the review paper? The authors have described MSC EVs for neurodegenerative disease which has been described multiple times in the literature. Moreover, there is no true evidence of how the EVs work and there are mixed results of their use between studies. Moreover, the authors have not focused enough on the diagnostic properties as mentioned in the title of the manuscript.
The first sentence of the introduction is extremely misleading, MSCs are not found in the brain or just few tiny exceptions and should not be used as an example for their presence in organs. Please review the literature and state the correct facts about MSCs and their origin and function in adults which would be the target of their clinical use, not necessarily on fetuses.
Line 19 – typo – “MSC cells” mean “mesenchymal stem cell cells”, it should be MSC or MSCs?
Line 72 – what is PK? Should it be PD?
Figure 1 is not very understandable, please revise the figure.
A detailed introduction to MSCs should be given.
Do exosomes and other EVs from different MSCs (based on source, surface markers and passage number) have different therapeutic effects? How do these variables affect the EVs? Are all the EVs same? This is not discussed in this review
Line 544 – Huntington’s disease
Conclusions: The review does not add any value to the currently existing literature. This reviewer rejects the manuscript in the current form.
Minor edits on the English languages should be performed
Author Response
REVIEWER 2
The manuscript entitled “Mesenchymal Stem Cell-Derived Extracellular Vesicles: An Emerging Diagnostic and Therapeutic Biomolecules for Neurodegenerative Disabilities” is a review article that has described different EVs and their potential therapeutic effects in neurodegenerative diseases. The manuscript has some major concerns. What is the novelty in the review paper? The authors have described MSC EVs for neurodegenerative disease which has been described multiple times in the literature. Moreover, there is no true evidence of how the EVs work and there are mixed results of their use between studies. Moreover, the authors have not focused enough on the diagnostic properties as mentioned in the title of the manuscript.
Response to reviewer comments
The idea of extracellular vesicles (EVs) and their possible therapeutic uses has been the subject of research fin the recent years. EVs play a vital role in facilitating cell-to-cell communication by transporting bioactive molecules like proteins, nucleic acids, and lipids. Among these, EVs derived from mesenchymal stem cells (MSCs) have drawn significant interest due to their regenerative and immunomodulatory capabilities, all while avoiding the potential risks associated with administering whole MSCs. Therefore, review articles are required to give continuous reviewing of the rapidly growing science and research in this field.
MSC-EVs have become a highly promising area of research with extensive therapeutic possibilities. In this review article, we tried to follow-up this rapidly evolving field and offer a comprehensive overview of current research, consolidate knowledge from various studies, and identify gaps in the research. Through critical analysis and comparative assessments, this review will help researchers understand key findings, advancements, and trends while pinpointing areas that require further investigation. Moreover, they elucidate the underlying mechanisms involved in therapeutic applications, significantly contributing to the translational impact of MSC-derived EVs research by bridging the gap between basic science and clinical applications.
We tried to improve our manuscript by adding several paragraphs about the recent advancement in biotechnology and bioengineering of MSC-derived EVs science and its applications in every diseases listed in this review. Please review the track changes file to see these changes. We also added Table 2 The miRNAs found in MSC-derived EVs, their role and applications. The Table provided new and informative applications for one of the most important component of EVs.
We agree with the reviewer that the exact mechanism of action of EVs is still a mystery. However, there are experimental and clinical evidences for the success of these agents in treating difficult situations, where the other tools fail to deliver any improvement.
We added a new section to specifically illustrate the diagnostic potential of MSCEVs as follows:
Table 3 provides an overview on the diagnostic applications of EVs derived from MSCs in various neurological disorders. It emphasizes the potential uses of MSC-EVs as diagnostic tools and highlights specific examples supporting their application. In AD, MSC-EVs present promising opportunities for diagnostics and biomarker identification. These EVs carry specific proteins and microRNAs that can be analyzed to identify early biomarkers of AD. Exosome-derived tau and amyloid-beta levels in CSF can indicate AD progression, while the detection of Aβ42 in CSF or blood aids in early diagnosis. Additionally, MSC-EVs can be utilized to track disease progression, monitor treatment response, and assess the effectiveness of therapeutic interventions in AD [1-5].
In PD, MSC-EVs also serve as potential diagnostics. Levels of α-synuclein and DJ-1 in MSC-EVs from CSF may indicate early signs of PD, and monitoring miR-34a levels in EVs can help track disease progression. Moreover, LRRK2-enriched EVs and miR-124-enriched EVs hold promise as potential biomarkers for PD detection [6-8].
In MS, quantification of myelin EVs in CSF shows promise as a potential biomarker for disease activity and progression. These myelin-derived EVs offer valuable insights into the disease's status and advancement [9]. For ALS, MSC-EVs could aid in early diagnosis and predicting disease outcomes by analyzing ALS-related proteins, including SOD1, TDP-43, pTDP-43, and FUS in these EVs [10]. Likewise, in HD, elevated total huntingtin levels in EVs from plasma of HD groups compared to controls suggest the potential use of huntingtin protein levels in EVs as a diagnostic biomarker for the disease [11].
These data highlight the immense diagnostic potential of MSC-EVs in neurological disorders, showcasing their capability to provide valuable information for early detection, disease monitoring, and treatment assessment.
Table 3. The diagnostic applications of MSC-EVs in neurological disorders.
|
Neurological Disorder |
Diagnostic Uses & Applications |
Examples |
reference |
|
Alzheimer's Disease |
Biomarker Identification: EVs derived from MSCs carry specific proteins, microRNAs, and other molecules that can be analyzed to identify early biomarkers of Alzheimer's disease. |
Exosome-derived tau and amyloid-beta levels in CSF can indicate Alzheimer's progression. |
[3,4] |
|
Detection of Aβ42 in CSF or blood to aid in early Alzheimer's diagnosis. |
Aβ42. |
[5] |
|
|
Tracking disease progression, monitoring treatment response, and early detection. |
Downregulation of miR-212 and miR-132-enriched EVs in AD |
[2] |
|
|
Assessing the effectiveness of therapeutic interventions |
EVs containing Tau protein |
[1] |
|
|
Parkinson's Disease |
Biomarker Detection: EVs released by MSCs carry specific proteins and microRNAs that can serve as biomarkers to detect early signs of Parkinson's disease. |
α-synuclein and DJ-1 levels in MSC-EVs from CSF as a potential biomarker for Parkinson's. |
[8] |
|
Monitoring disease progression by assessing miR-34a levels in EVs. |
miR-34a over-expression in PD |
[7] |
|
|
biomarkers to detect PD. |
miR-124-enriched EVs in PD |
[6] |
|
|
Multiple Sclerosis |
Quantification of myelin EVs in CSF as a potential biomarker for disease activity and progression. |
Myelin-derived EVs |
[9] |
|
Amyotrophic Lateral Sclerosis |
Early diagnosis and predicting disease outcome. |
ALS-related proteins, including SOD1, TDP-43, pTDP-43 and FUS |
[10] |
|
Huntington's Disease |
Elevated total huntingtin levels in EVs from plasma of HD groups compared to controls |
Huntingtin protein |
[11] |
The first sentence of the introduction is extremely misleading, MSCs are not found in the brain or just few tiny exceptions and should not be used as an example for their presence in organs. Please review the literature and state the correct facts about MSCs and their origin and function in adults which would be the target of their clinical use, not necessarily on fetuses.
Response to reviewer comments
The first sentence was modified as follows”
“Mesenchymal stem cells (MSCs) are a type of adult stem cell with the ability to develop into different types of mesoderm-derived cells. Traditionally, MSCs were not considered to be naturally present in the brain; however, recent studies have indicated that they might exist as perivascular cells in nearly all adult tissues, including the brain [12].”
Line 19 – typo – “MSC cells” mean “mesenchymal stem cell cells”, it should be MSC or MSCs?
Response to reviewer comments
Corrected
Line 72 – what is PK? Should it be PD?
Response to reviewer comments
Corrected
Figure 1 is not very understandable, please revise the figure.
Response to reviewer comments
Figure 1 is modified to be more clear.
A detailed introduction to MSCs should be given.
Response to reviewer comments
The introduction is modified to give more details about MScs.
Do exosomes and other EVs from different MSCs (based on source, surface markers and passage number) have different therapeutic effects? How do these variables affect the EVs?
Response to reviewer comments
Exosomes and other extracellular vesicles (EVs) originating from various Mesenchymal Stem Cells (MSCs) can display diverse therapeutic effects based on factors like their origin, surface markers, and passage number. These variables significantly influence the composition and functional characteristics of EVs, thereby impacting their potential for therapeutic applications.
It is important to acknowledge that while these variables contribute to the diversity of MSC-derived EVs, they can also offer advantages. The varied cargo and properties of EVs enable researchers to customize and fine-tune EV-based therapies for specific diseases and applications. However, standardizing EV preparations for clinical use poses challenges, and further research is necessary to fully comprehend and harness the therapeutic potential of MSC-derived EVs. As EV research advances, more insights into how these variables influence therapeutic effects are expected, paving the way for more precise and effective regenerative medicine and disease treatments.
Based on these, we added a new section to describe the differences and challenges facing MSC-EVs and its technology and bioengineering. Please review the track changes file.
Are all the EVs same? This is not discussed in this review
Response to reviewer comments
Not all extracellular vesicles (EVs) possess identical characteristics. These are a diverse array of small membrane-bound structures discharged by cells into the extracellular environment, serving vital functions in cell-to-cell communication and facilitating the transfer of various molecules, including proteins, nucleic acids (such as RNA and DNA), and lipids, among cells.
Although both exosomes and microvesicles participate in intercellular communication, they exhibit distinct compositions and biogenesis mechanisms. Moreover, their cargo, surface markers, and functions can vary depending on the cell type, physiological condition, and external stimuli. Furthermore, even within these broader categories, there exists additional heterogeneity. EVs originating from different cell types, tissues, and conditions may possess unique cargo compositions and functions. Ongoing research in this field continues to unveil the intricacy and diversity of extracellular vesicles and their significance in various physiological and pathological processes.
We modified and updated section 2 to cover this point.
Line 544 – Huntington’s disease
Response to reviewer comments
Corrected
References
- Bell, B.J.; Malvankar, M.M.; Tallon, C.; Slusher, B.S. Sowing the seeds of discovery: tau-propagation models of Alzheimer’s disease. ACS chemical neuroscience 2020, 11, 3499-3509.
- Cha, D.J.; Mengel, D.; Mustapic, M.; Liu, W.; Selkoe, D.J.; Kapogiannis, D.; Galasko, D.; Rissman, R.A.; Bennett, D.A.; Walsh, D.M. miR-212 and miR-132 Are Downregulated in Neurally Derived Plasma Exosomes of Alzheimer's Patients. Frontiers in neuroscience 2019, 13, 1208, doi:10.3389/fnins.2019.01208.
- Guix, F.X.; Corbett, G.T.; Cha, D.J.; Mustapic, M.; Liu, W.; Mengel, D.; Chen, Z.; Aikawa, E.; Young-Pearse, T.; Kapogiannis, D. Detection of aggregation-competent tau in neuron-derived extracellular vesicles. International journal of molecular sciences 2018, 19, 663.
- Hampel, H.; Bürger, K.; Teipel, S.J.; Bokde, A.L.; Zetterberg, H.; Blennow, K. Core candidate neurochemical and imaging biomarkers of Alzheimer's disease. Alzheimer's & dementia : the journal of the Alzheimer's Association 2008, 4, 38-48, doi:10.1016/j.jalz.2007.08.006.
- Molinuevo, J.L.; Ayton, S.; Batrla, R.; Bednar, M.M.; Bittner, T.; Cummings, J.; Fagan, A.M.; Hampel, H.; Mielke, M.M.; Mikulskis, A. Current state of Alzheimer’s fluid biomarkers. Acta neuropathologica 2018, 136, 821-853.
- Angelopoulou, E.; Paudel, Y.N.; Piperi, C. miR-124 and Parkinson’s disease: A biomarker with therapeutic potential. Pharmacological research 2019, 150, 104515.
- Grossi, I.; Radeghieri, A.; Paolini, L.; Porrini, V.; Pilotto, A.; Padovani, A.; Marengoni, A.; Barbon, A.; Bellucci, A.; Pizzi, M. MicroRNA‑34a‑5p expression in the plasma and in its extracellular vesicle fractions in subjects with Parkinson's disease: An exploratory study. International Journal of Molecular Medicine 2021, 47, 533-546.
- Hong, Z.; Shi, M.; Chung, K.A.; Quinn, J.F.; Peskind, E.R.; Galasko, D.; Jankovic, J.; Zabetian, C.P.; Leverenz, J.B.; Baird, G. DJ-1 and α-synuclein in human cerebrospinal fluid as biomarkers of Parkinson’s disease. Brain 2010, 133, 713-726.
- Jagot, F.; Davoust, N. Is it worth considering circulating microRNAs in multiple sclerosis? Frontiers in immunology 2016, 7, 129.
- Gagliardi, D.; Bresolin, N.; Comi, G.P.; Corti, S. Extracellular vesicles and amyotrophic lateral sclerosis: from misfolded protein vehicles to promising clinical biomarkers. Cellular and molecular life sciences : CMLS 2021, 78, 561-572, doi:10.1007/s00018-020-03619-3.
- Ananbeh, H.; Novak, J.; Juhas, S.; Juhasova, J.; Klempir, J.; Doleckova, K.; Rysankova, I.; Turnovcova, K.; Hanus, J.; Hansikova, H. Huntingtin co-isolates with small extracellular vesicles from blood plasma of TgHD and KI-HD pig models of Huntington’s disease and human blood plasma. International journal of molecular sciences 2022, 23, 5598.
- Appaix, F.; Nissou, M.F.; van der Sanden, B.; Dreyfus, M.; Berger, F.; Issartel, J.P.; Wion, D. Brain mesenchymal stem cells: The other stem cells of the brain? World journal of stem cells 2014, 6, 134-143, doi:10.4252/wjsc.v6.i2.134.

Reviewer 3 Report
This review manuscript summarized the three types of extracellular vesicles (EVs) and then mainly discussed the effects of mesenchymal stem cells (MSCs)-derived EVs in treating several neurodegenerative diseases, including Alzheimer's disease (AD), multiple sclerosis (MS), Parkinson's disease (PD), amyotrophic lateral sclerosis (ALS), and Huntington's disease (HD). To make it more interesting to ordinary readers and the researchers who work in the field, the manuscript should also include following topics and each can be one section with a subtitle:
1. Why MSCs but not other types stem cells have become so popular in treating the neurological diseases. The authors should state why MSCs and their derived EVs are so useful and intriguing to many researchers.
2. MSC EV extraction procedures; What is the most popular isolation methods for MSC-derived EVs? What caution should be taken in preparation of MSC cultures and EV isolation?
3. Please also cover the “recent biotechnology and engineering research to modify the contents and application of EVs in treating specific neurological diseases”.
4. Please include the Challenge and Future Perspectives, which can be a separated section from Conclusion.
The English is OK.
Author Response
REVIEWER 3
This review manuscript summarized the three types of extracellular vesicles (EVs) and then mainly discussed the effects of mesenchymal stem cells (MSCs)-derived EVs in treating several neurodegenerative diseases, including Alzheimer's disease (AD), multiple sclerosis (MS), Parkinson's disease (PD), amyotrophic lateral sclerosis (ALS), and Huntington's disease (HD). To make it more interesting to ordinary readers and the researchers who work in the field, the manuscript should also include following topics and each can be one section with a subtitle:
Response to reviewer comments
We express our gratitude to the reviewer for bringing up significant points, and we made every effort to address them adequately in our response.
Why MSCs but not other types stem cells have become so popular in treating the neurological diseases. The authors should state why MSCs and their derived EVs are so useful and intriguing to many researchers.
Response to reviewer comments
Mesenchymal stem cells (MSCs) have become increasingly popular for treating neurological diseases due to several key reasons:
1.Abundant Source: MSCs are readily obtainable from various tissues, such as bone marrow, adipose tissue, umbilical cord, and placenta. This widespread availability makes them highly practical for both research and clinical applications.
- Low Immunogenicity: MSCs exhibit low immunogenicity, reducing the likelihood of eliciting a severe immune response when transplanted into patients.
- Anti-inflammatory Properties: MSCs possess powerful anti-inflammatory and immunomodulatory capabilities. They can secrete numerous factors that suppress inflammation and regulate the immune response, which proves beneficial in neuroinflammatory conditions like multiple sclerosis or stroke.
- Trophic Factor Secretion: MSCs release a variety of trophic factors that promote cell survival, tissue repair, and neurogenesis.
- Ability to Cross Blood-Brain Barrier: MSCs have demonstrated the ability to cross the blood-brain barrier, enabling them to reach the central nervous system.
- Safe and Well-Tolerated: Clinical studies have shown that MSC transplantation is generally safe and well-tolerated in patients, with a low risk of adverse effects, making them a promising option for neurological disease treatments.
- Preclinical Success: MSCs have shown promising results in preclinical studies, particularly in animal models of various neurological diseases. These studies have provided compelling evidence of their therapeutic potential, paving the way for clinical trials.
Based on these factors we sought to review the MSCs EVs.
We also modified the text and added new paragraph to imply this (The second paragraph in the introduction) as follows:
“The MSCs have become a promising therapeutic option for treating neurological diseases due to their distinctive characteristics [5]. These cells are plentiful and easily obtainable from various tissues, and they exhibit low immunogenicity, allowing for allogeneic transplantation without triggering significant immune responses. Additionally, MSCs possess potent anti-inflammatory and immunomodulatory properties, which make them well-suited for addressing neuroinflammatory conditions [6]. Through the secretion of trophic factors [7], they actively promote tissue repair, neurogenesis, and neuronal survival. Their unique ability to cross the blood-brain barrier facilitates delivery to the central nervous system. Notably, clinical studies have demonstrated that MSC transplantation is well-tolerated and associated with low adverse effects [8].”
MSC EV extraction procedures; What is the most popular isolation methods for MSC-derived EVs? What caution should be taken in preparation of MSC cultures and EV isolation?
Response to reviewer comments
we thank the reviewer for this comment.
We added two paragraphs at the end of section 2 to cover these points as follows:
“The standardization and extraction of MSCs-EVs revealed various characteristics. The most popular isolation methods for MSC-derived EVs were based on the size of EV and include ultracentrifugation, differential ultracentrifugation, density gradient ultracentrifugation and ultrafiltration [29]. A larger yield is obtained while maintaining the biophysical and functional characteristics of the EV isolated from stem cell culture using ultrafiltration followed by size exclusion chromatography (SEC) [30]. Other methods were also used, comprising immune affinity capture. However, it was less efficient than SEC [31].
Extracting EVs from MSC cultures is a critical process in biomedical research, and it requires meticulous attention to multiple precautions to ensure the reliability and quality of outcomes. Maintaining a sterile environment throughout the culture and isolation process is crucial to prevent contamination, while the quality and characteristics of the MSCs used can significantly influence the composition and function of the isolated EVs. Therefore, it is essential to utilize high-quality MSCs that meet specific criteria. Additionally, proper handling of samples is of utmost importance to preserve the integrity and stability of EVs. This involves careful collection, storage, and transportation to minimize degradation or loss. To enhance the yield and purity of EVs, optimizing the isolation method is necessary, which may include adjusting centrifugation parameters based on study requirements and EV characteristics. Lastly, to ensure the quality and purity of the isolated EVs, thorough characterization using techniques like electron microscopy, nanoparticle tracking analysis, and protein analysis is indispensable [32,33]. These measures collectively contribute to the success of MSC culture and EV isolation, advancing the understanding and potential applications of EVs in various biomedical fields with reliable and meaningful results.”
Please also cover the “recent biotechnology and engineering research to modify the contents and application of EVs in treating specific neurological diseases”.
Response to reviewer comments
We thank the reviewer for thisnimportant comment.
We now added summary of the recent work in biotechnology and engineering of MSC-EVs and added it to every neurological discorder in the manuscript.
In AD we added the following
“Although MSC-EVs hold promise for treating AD, their full potential is hindered by several challenges. These issues include inadequate targeting efficiency, inconsistent treatment results, and limited production yield [128]. To overcome these limitations and enhance the effectiveness of MSC-EVs as AD treatments, it is essential to functionalize and engineer the EVs structures. Various methods can be employed for this purpose, such as preconditioning the parental cells to improve the natural treatment's effectiveness, incorporating therapeutic cargo or drug loading into MSC-EVs, modifying the surface of MSC-EVs to enhance targeting capabilities, and utilizing artificial MSC-EVs to scale up production [129].
MSC-EV-based AD treatments were accomplished using different types of mesenchymal stem cells: bone marrow-derived MSCs (mBMSCs), adipose-derived MSCs (mADSCs), and human umbilical cord-derived MSCs (hUCMSCs). To enhance the therapeutic efficacy of MSC-EVs, they are modified with specific peptides like the rabies virus glycoprotein (RVG) peptide or loaded with miRNAs like miR-29 and miR-22 [129]. MSC-EVs were administered through various routes, including intravenous (IV), intracranial injection (IN), and dorsal hippocampus injection. IV administration of RVG-modified MSC-EVs has shown promise in improving learning and memory by reducing Aβ deposition and astrocyte activation while promoting the production of anti-inflammatory factors. Similarly, dorsal hippocampus injection of miR-29-loaded MSC-EVs reduces BACE1 expression and activates PKA/CREB, leading to improved cognitive function [130]. Preconditioning MSCs in an AD environment before administration is another promising strategy. Hypoxic preconditioning of mADSCs shifts microglial M1/M2 polarization, reduces inflammatory factors, and upregulates TREM2 expression, thereby improving cognitive function [131].”
In PD, we added the following paragraph
“One of the major obstacles of PD treatment is the access to the damaged cells. However, recent bioengineering research had led to production of genetically modified cells with improved therapeutic efficiency. Improved adhesion, migration and survival are new methods for not only preserving but also increasing the biological properties and therapeutic potential of MSCs [165]. MSCs that have been genetically designed to produce specific neurotrophic factors, such as brain-derived neurotrophic factor, or MSCs that have been modified to boost their survival and ability to migrate towards the lesion location. Concerning PD, various studies have utilized engineered MSCs expressing vascular endothelial growth factor, tyrosine hydroxylase, or modified to enhance the production of cerebral dopamine neurotrophic factor or glial cell-derived neurotrophic factor. These experiments have shown promising outcomes in preclinical rodent models [166]. Furthermore, genetically engineered EVs showed promising results in PD animal models. When subjected to catalase-loaded EVs in a cell culture environment, macrophages that were activated with lipopolysaccharide (LPS) and tumor necrosis factor (TNF) demonstrated decreased levels of reactive oxygen species (ROS). In a mouse model of PD using 6-OHDA, administering catalase-loaded EVs resulted in reduced activation of microglia compared to the application of free catalase. These findings suggest that delivering catalase through EVs has the potential to effectively decrease oxidative stress and neuroinflammation in PD [167]. Interestingly, the introduction of dopamine-loaded EVs led to a remarkable 15-fold increase in dopamine distribution within the brain. This heightened distribution not only resulted in improved therapeutic effectiveness but also significantly reduced systemic toxicity compared to the administration of free dopamine. These findings suggest that utilizing EV-based drug delivery holds promising potential as a viable and effective treatment option for PD [168].”
Please include the Challenge and Future Perspectives, which can be a separated section from Conclusion.
Response to reviewer comments
The text is modified to include the new section of the challenges and as follows:
“10. Challenge and Future Perspectives
Caution should be exercised when interpreting studies comparing the effectiveness of MSC-EVs and their cellular counterparts due to limitations in current methods of EV quantification [202]. Furthermore, there is insufficient evidence regarding the optimal therapeutic dosage of EVs and their long-term effects [203]. Some types of tissue damage may not require multiple administrations of MSC-EVs, while others may necessitate repeated treatments, which can increase the burden and expenses for the patient. Addressing the challenge of heterogeneity among MSC-EVs is crucial before their widespread clinical application. Different types of MSC-EVs have varying levels of treatment efficacy, underscoring the need for further research to identify subgroups with the greatest therapeutic potential. Improved methodologies and research in genomics, proteomics, and other fields are necessary to accurately classify and differentiate EV subtypes. Additionally, it is important to enhance the reproducibility of large-scale EV production with high purity and specific therapeutic activity. This calls for the development of robust in vitro quality control systems tailored to the specific requirements of EV-based treatments.
The MSC-EVs have emerged as a promising area of research in regenerative medicine and therapeutic applications. These tiny membrane-bound vesicles carry bioactive molecules and can influence the behavior of target cells and tissues. Despite their potential, MSC-EVs face challenges in standardization, scalability, payload loading, storage, and targeting. However, the future prospects for MSC-EVs are exciting, with potential applications in treating various diseases, tissue regeneration, immunomodulation, personalized medicine, drug delivery, and as biomarkers for diagnostics. The study of MSC-EVs has also advanced nanomedicine and drug delivery systems. Ongoing research and clinical trials may lead to the approval of MSC-EV-based therapies and contribute to significant advancements in the field of regenerative medicine and therapeutics.
- Conclusions
The therapeutic potential of MSC-EVs has been demonstrated in various cases of tissue injury. In vivo studies have shown that MSC-EVs are as effective as the parent cells in promoting tissue regeneration. This finding is intriguing for potential clinical use of MSC-EVs, suggesting that they could be a cost-effective alternative to MSC-based therapies. Current preclinical research suggests that MSC-EVs directly target macrophages and the injured tissue, indicating that their positive impact on tissue regeneration is achieved through modulation of immune responses and direct interaction with the tissue. A deeper understanding of this combined activity of MSC-EVs can facilitate better targeting of specific tissues and enhance treatment efficacy. Importantly, uncovering the mechanism of action of MSC-EVs is crucial for determining their legal status and maximizing their therapeutic effects, which are vital considerations for their potential clinical applications.”

Round 2
Reviewer 2 Report
The authors have made the necessary changes, however most of the newly added contents do not have references and citations. Please add up-to-date references and citations.
Minor spell check is required.
Author Response
REVIEWER 2
Comments and Suggestions for Authors
The authors have made the necessary changes; however, most of the newly added contents do not have references and citations. Please add up-to-date references and citations.
Response to reviewer comments
We updated the references and citations, please check the track changes file.
Comments on the Quality of English Language
A minor spell check is required.
Response to reviewer comments
The manuscript is now double-checked for language issues and spelling.
Reviewer 3 Report
All my concerns have been addressed.
Author Response
REVIEWER 3
Comments and Suggestions for Authors
All my concerns have been addressed.
Response to reviewer comments
We have updated the manuscript to include article citations as well as English language proof reading.